# Pharmacological Blockade of TGF-Beta Reduces Renal Interstitial Fibrosis in a Chronic Ischemia–Reperfusion Animal Model

Zesergio Melo [1],* , Julio Palomino [2], Adriana Franco-Acevedo [3], David García [2], Ricardo González-González [2], Maritza G. Verdugo-Molinares [4], Eliseo Portilla-de Buen [2], Bibiana Moreno-Carranza [5], Clotilde Fuentes-Orozco [6], Francisco J. Barbosa-Camacho [6], Emilio A. Reyes-Elizalde [6], Laura Cortés-Sanabria [6] and Alejandro González-Ojeda [6],*

[1] CONACYT-Centro de Investigación Biomédica de Occidente, Instituto Mexicano del Seguro Social, Guadalajara 44340, Mexico
[2] Centro de Investigación Biomédica de Occidente, Instituto Mexicano del Seguro Social, Guadalajara 44340, Mexico; juliopalominop8@gmail.com (J.P.); davidga28@gmail.com (D.G.); qfbragg26@gmail.com (R.G.-G.); dreportilla@gmail.com (E.P.-d.B.)
[3] Department of Pathology and Laboratory Medicine, University of California, Los Angeles, CA 90095, USA; ady_francoa@hotmail.com
[4] Maestría en Ciencias en Innovación Biotecnológica, Centro de Investigación y Asistencia en Tecnología y Diseño del Estado de Jalisco, Guadalajara 44270, Mexico; maverdugo_al@ciatej.edu.mx
[5] Escuela de Medicina, Universidad Anáhuac Querétaro, Queretaro 76246, Mexico; bibianamorenocarranza@gmail.com
[6] Unidad de Investigación Biomédica 02, Hospital de Especialidades, Centro Médico Nacional de Occidente, Instituto Mexicano del Seguro Social, Guadalajara 44349, Mexico; clotilde.fuentes@gmail.com (C.F.-O.); efebeka_@hotmail.com (F.J.B.-C.); dreyes.emilio@gmail.com (E.A.R.-E.); cortes_sanabria@yahoo.com.mx (L.C.-S.)
* Correspondence: zcmelo@conacyt.mx (Z.M.); avygail5@gmail.com (A.G.-O.); Tel.: +52-3331294165 (A.G.-O.)

**Abstract:** The targeting of transforming growth factor β (TGF-β) has been shown to reduce complications related to ischemia-reperfusion injury (IRI) post-surgically. Pirfenidone (PFD) specifically inhibits TGF-β expression and has been demonstrated to provide protection from IRI in short-term allograft models, though not yet in long-term models. A chronic unilateral IRI model was established using male Wistar rats. The animals were divided into two groups: one with IRI and a pre-treatment of PFD (0.5 mg/kg) followed by 0.5 mg/kg/day of orally administered PFD for 30 days, and a control group without PFD treatment. A sham group was also included. Kidneys and blood samples were collected after 30 days, and the renal function was evaluated by measuring the serum creatinine and KIM-1 levels. RT-PCR was used to analyze fibrosis-related genes, and Luminex to quantify the pro-inflammatory serum IL-18 cytokine. Renal section staining and histological analysis were used to detect collagen deposits. Comparison within the groups showed an increase in serum creatinine and KIM-1 expression after IRI in the control group, while PFD reduced COLL1A1 and TGF-β expression and demonstrated a reduction in fibrosis through histological stains. The treatment group also showed a reduction in IL-18. Our results suggest that PFD exerts protective effects on chronic renal IRI, reducing fibrosis development and inflammation. This study provides new insights into the treatment and management of chronic renal function loss after IRI.

**Keywords:** pirfenidone; transforming growth factor β; interstitial fibrosis; renal ischemia–reperfusion; chronic kidney disease

## 1. Introduction

A rapid reduction in kidney function may progress to chronic kidney disease. Nowadays, there is no curative treatment for patients with end-stage chronic kidney disease, and the only remaining therapeutic option is kidney transplantation [1]. Nevertheless,

surgical conditions such as the interruption of blood flow as well as of oxygen perfusion may cause newly transplanted organs to suffer ischemia–reperfusion injury (IRI) following the restoration of blood flow [2]. Renal IRI commonly associates with a high prevalence of fibrosis development and graft rejection [3]. The more frequent causes of progression are dysregulated tubular repair, interstitial tubule inflammation, exacerbated proliferation of fibroblasts, and extracellular matrix deposition [4–6]. These processes are known to promote interstitial fibrosis and decrease kidney functions. Additionally, post-ischemic fibrosis is associated with the presence of inflammation and long-term microvessel loss, causing serious injury to epithelial and endothelial cells [7].

Transforming Growth Factor β (TGF-β) is a multifunctional cytokine that plays an important role in many biological processes. In the context of fibrosis, TGF-β acts as a profibrotic mediator and promotes the activation and differentiation of fibroblasts, which may result in the overproduction of extracellular matrix components in renal tissues [8]. Progressive fibrosis of the renal interstitium, tubules and blood vessels is associated with kidney transplant rejection [9]. In this regard, fibrosis serves as a prognostic marker of renal allograft failure and rejection [9]. Therefore, it is crucial to develop and assess pharmacological strategies to reduce renal fibrosis caused by IRI [10]. Multiple studies demonstrate that targeting TGF-β reduces fibrosis in kidney disease and improve the renal function [11,12].

Pirfenidone (PFD) is a type of 2-pyridone molecule with phenyl and methyl substituents at positions 1 and 5, respectively (see Figure 1). Preclinical studies have shown that PDF antifibrotic activity is mediated by inhibition of TGF-β expression [13]. Additionally, PFD has shown to be effective and safe in treating idiopathic pulmonary fibrosis and some kidney diseases [14,15]. Currently, PFD is not widely used in the treatment of renal allograft failure or rejection. However, evidence supports the antifibrotic and anti-inflammatory activities of PFD in the kidney [16]. Clinical studies have shown that PFD can improve the glomerular filtration rate in patients with diabetic nephropathy and focal and segmental glomerulosclerosis [13,17]. Moreover, an animal study showed that PFD protects the kidney against acute IRI through its antioxidant and anti-inflammatory properties [18]. In addition, more in vivo results suggest that PFD decreases renal fibrosis in a chronic renal allograft model [19]. Despite this evidence, the role of PFD in protecting the kidney from long-term IRI-induced fibrosis has not been exhaustively explored.

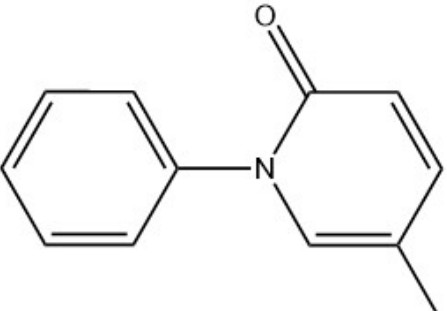

**Figure 1.** Chemical structure of pirfenidone. Figure done using ChemDraw, PerkinElmer Informatics, Waltham, MA, USA.

In the present study, we aimed to determine whether PFD could reduce fibrosis in a chronic IRI rat model. We also examined the effect of PFD on the expression of profibrotic molecules and collagen deposition in renal tissues, in order to gain insight into its mechanism of action. Our results provide new evidence that PFD may be a promising therapeutic agent for renal fibrosis.

## 2. Results

### 2.1. PFD Improves the Renal Function and Affects Weight Gain

The effect of PFD on the renal function was analyzed by measuring serum creatinine levels and KIM-1 mRNA expression in rats that underwent sham surgery, IRI, or IRI plus daily PFD treatment until day 30. The results confirmed the increase of serum creatinine (IRI vs. IRI + PFD, $p = 0.039$) and KIM-1 renal mRNA expression after IRI (Figure 2A,B, black bar); however, PFD treatment significantly reduced them (Figure 2A,B, gray bar). Additionally, we analyzed the body weight of the experimental animals at the beginning and at the end of 30 days of treatment. We found that the animals in the sham and IRI groups showed a significantly different body weight gain of 64.3 ± 8.67 and 73.6 ± 7.68 g, respectively ($p < 0.05$). However, the group that received the PFD treatment showed a poor weight gain of only 30.4 ± 13.79 g, with no statistical difference between the two measurements, as shown Figure 2C.

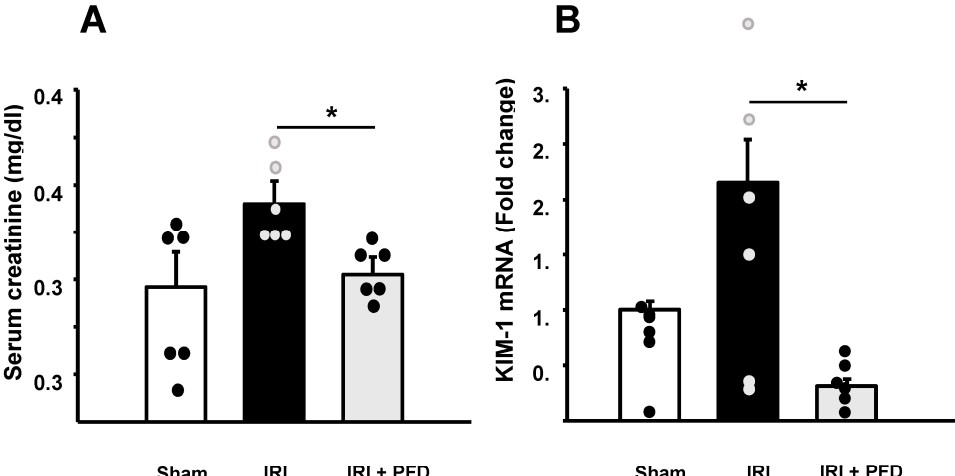

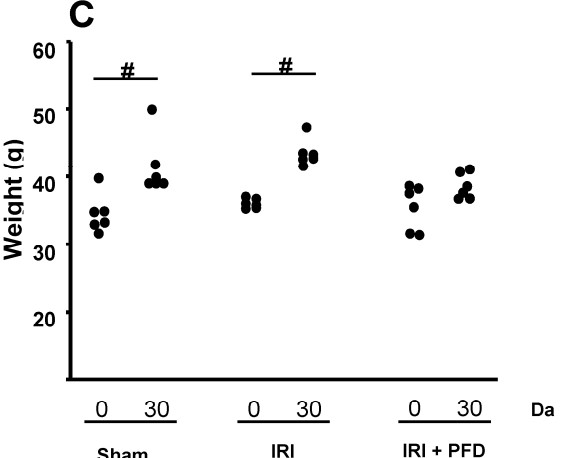

**Figure 2.** PFD reduces IRI. (**A**). Creatinine level was assessed by the dry chemistry method. (**B**). Intrarenal expression of KIM-1 was evaluated by quantitative RT-PCR. Hprt was used as housekeeping gene for normalization. (**C**). The animals' weight was evaluated before surgery and at the end of the 30 days of reperfusion; the means are represented by dots. The sham group is represented by white bars, the IRI group by black bars, and the IRI + PFD group by light gray bars. Dots represents each animal analyzed. Values are means ± SE ($n = 6$). * = $p < 0.05$ vs. IRI. # = $p < 0.05$ vs. 30 days after surgery.

## 2.2. PFD Reduces the Expression of Col1A1

To explore how PFD could influence the fibrotic response after renal IRI, we assessed Col1A1 and TGF-beta mRNA expression levels in kidney tissues. The RT-PCR analysis showed a significant increase in the expression of Col1A1 gene 30 days after IRI. Additionally, we observed a reduction in the expression of Col1A1 in animals under PFD treatment (Figure 3A). Although there appeared to be a reduction in TGF-beta expression after PFD treatment, the statistical analysis showed no significant difference (Figure 3B).

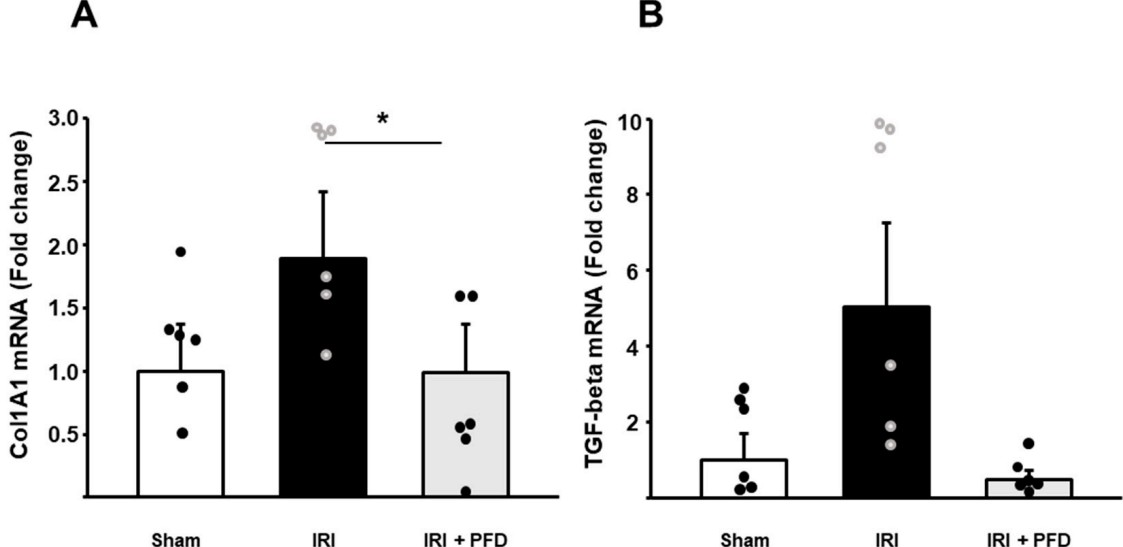

**Figure 3.** Pirfenidone reduces fibrosis-related genes' expression. (**A**). Col1A1 and (**B**). TGF β mRNA expression was analyzed by real-time PCR. The sham group is represented by white bars, the IRI group by black bar, and the IRI + Pirfenidone group by light gray bar. Dots represents each animal analyzed. Values are means ± SE ($n = 6$). * $p < 0.05$ vs. IRI.

## 2.3. PFD Improves Kidney Interstitial Fibrosis after IRI

To determine the level of interstitial fibrosis in experimental animals' kidney tissues, we performed four commonly used histological stains: H&E, PAS, Masson's trichrome, and Sirius red (Figure 4A–D). According to our results, the sham animals did not present collagen deposits in the interstitial region of kidney tissues, showing a normal morphology without noticeable pathological changes. However, a significant accumulation of collagen fibrils was observed in the IRI group. Importantly, the PAS, Mason's, and Sirius red stains revealed intensely colored areas in the IRI group significantly reduced in the kidneys from the PFD group (Figure 4B–D). When we quantified the areas corresponding to collagen fibers stained in blue in Masson's trichrome-stained tissues (Figure 4C), we found that the IRI group had a renal interstitial fibrosis area index of 9.810 ± 1.45% compared to that of 3.027 ± 0.38% in the sham group (Figure 4E). In addition, PFD treatment significantly reduced the areas infiltrated by collagen fibrils after IRI, resulting in a renal interstitial fibrosis area index of 4.939 ± 0.378% (Figure 4E).

## 2.4. PFD Modulates Circulating IL-18 Concentration

To examine the influence of PFD on inflammatory mediators of IRI, we quantified the circulating IL-18 concentration as a marker of immune stimulation. As anticipated, IRI increased the circulating levels of IL-18 cytokine. Nonetheless, PFD treatment significantly reduced the concentration of IL-18 to 157.51 ± 58.436 pg/mL after IRI (Figure 5). This difference was significant ($p < 0.001$).

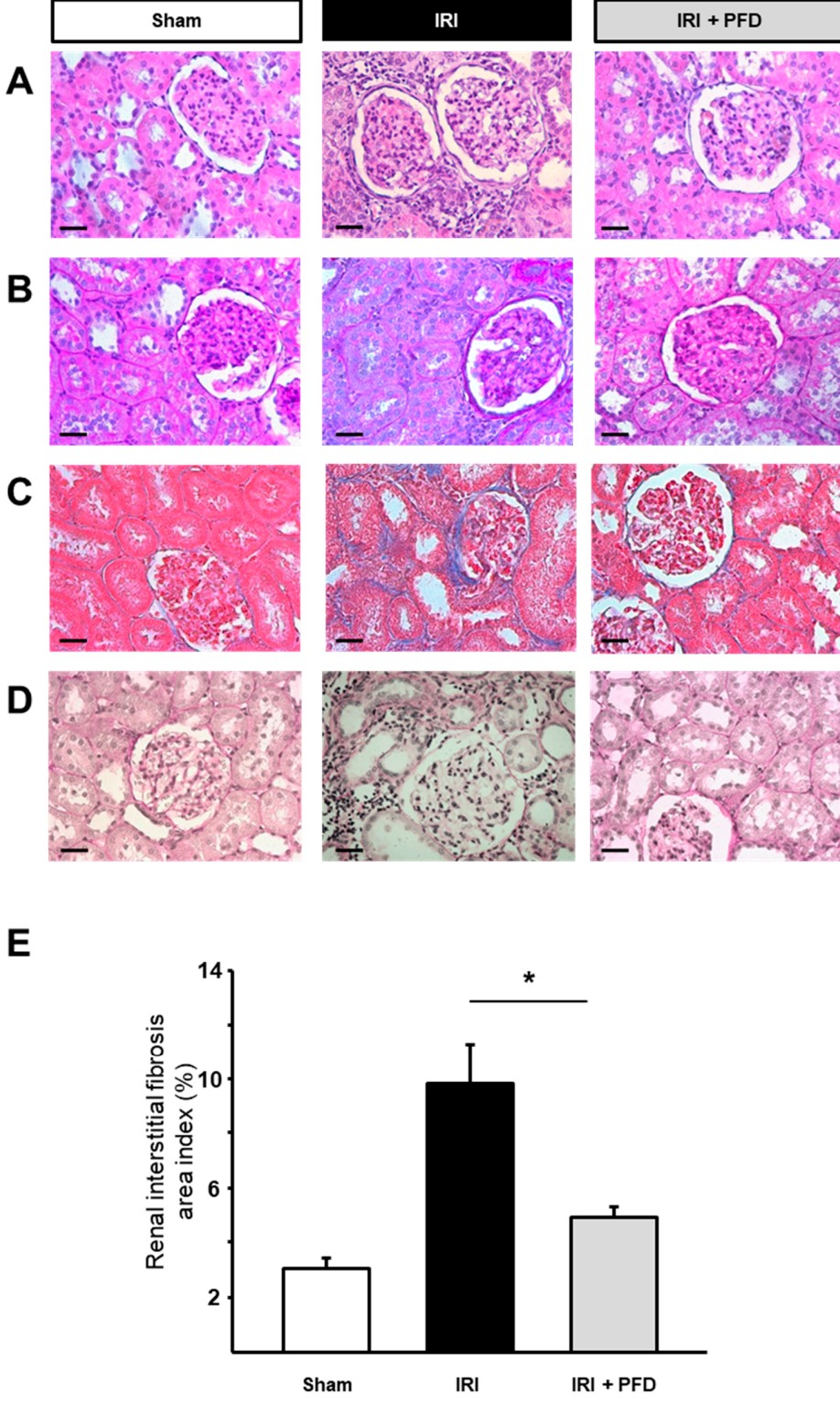

**Figure 4.** PFD reduces IRI-induced fibrosis after 30 days of treatment. Paraffin-embedded kidney sections were stained using (**A**). Hematoxylin and eosin (H&E), (**B**). Periodic acid–Schiff (PAS) stain, (**C**). Masson's trichrome, and (**D**). Sirius red. Pictures are representative. (**E**). Collagen deposition was quantified in ten pictures. The blue areas in Masson's trichrome were quantified using the ImageJ application. He sham group is represented by white bars, the IRI group by black bars, and the IRI + PFD group by light gray bars. Values are means ± SE (*n* = 10/animal). Scale bars represent 50 μm. * *p* < 0.05 vs. IRI.

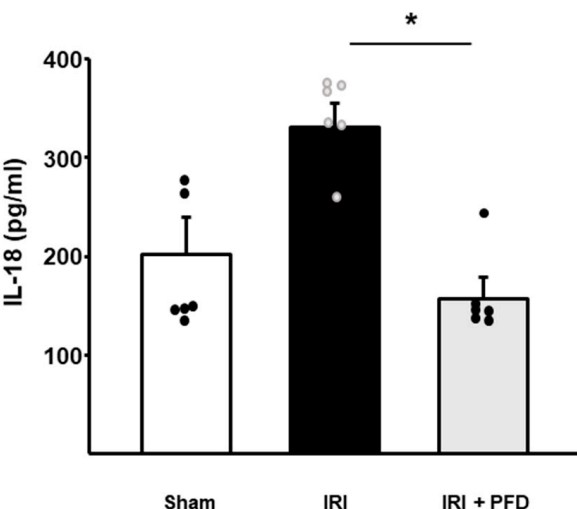

**Figure 5.** PFD modulates inflammatory responses. Luminex measurements of the inflammatory cytokine IL-18. Comparison of the concentration of circulating IL-18 cytokine in all three study groups. The sham group is represented by white bars, the IRI group by black bars, and the IRI + PFD group by light gray bars. Dots represents each animal analyzed. Values are means $\pm$ SE ($n$ = 6). * $p < 0.05$ vs. IRI.

## 3. Discussion

Our study demonstrated the protection mediated by PFD towards renal function by preventing long-term fibrosis development and kidney injury progression. We hypothesized that PFD administration after IRI would improve the renal function by modulating adaptive repair responses due to its antifibrotic activity. We found that PFD preserved the renal function, reduced the expression of ColA1 and TGF-beta, decreased fibrosis, and modulated inflammatory responses in an experimental model of chronic IRI.

PFD is an orally bioavailable drug with proven antifibrotic, antioxidant, and anti-inflammatory properties [20]. Although PFD has been widely used to reduce pulmonary fibrosis [21], its effects remain largely unexplored in the renal field. Previous studies reported that PFD is a renoprotective agent that exerts its functions partly by inhibiting the TGF-beta signaling pathway [22,23]. Some studies also suggested that pirfenidone is a promising therapeutic agent for individuals with diabetic nephropathy [24]. However, PFD's molecular mechanisms of action are still under study. Knowing the effects of PFD on the kidney opens the door to new pharmacological strategies for treating kidney IRI. In our study, we focused on exploring the antifibrotic effects of PFD. In addition to its ability as a therapeutic agent for fibrotic diseases, PFD has also shown promise in treating other conditions, such as cancer and inflammatory disorders [25]. Recent studies reported that PFD has anti-tumor effects and can inhibit the growth of cancer cell lines [26]. Additionally, PFD has been shown to have anti-inflammatory effects and has been used to treat inflammatory disorders such as rheumatoid arthritis [27]. These findings suggest that PFD has a broad range of therapeutic applications, and further research is needed to fully explore its therapeutic potential in different diseases. Overall, our study contributes to the growing body of evidence of the potential of PFD as a therapeutic option for fibrotic diseases, including renal fibrosis.

Most reports showing antifibrotic effects in the kidney were based on the Unilateral Ureteral Obstruction (UUO) model that rapidly and progressively induces a state of tubulointerstitial fibrosis in the affected kidney [28]. A dramatic decrease in profibrotic signals was observed in this model when PFD was used as a prophylactic [16]. Thus, with the administration of PFD in the long-term UUO model and acute IRI model, Lima-Posada et al. demonstrated a notable improvement in the renal function in just 24 h after the surgical intervention [18]. Our study found that the renal function remained at optimal levels despite IRI due to PFD treatment. We performed creatinine and KIM-1 measurements

30 days after surgery and saw an improvement in the levels of these markers, influenced by PFD.

Furthermore, our study provides insight into the potential therapeutic effects of PFD in a chronic IRI rat model, which more closely resembles the clinical scenario in human patients. The chronic IRI model induces renal fibrosis gradually over time, which better mimics the natural progression of the disease. The positive results seen in our study suggest that PFD may be effective in reducing fibrosis and improving the renal function in patients with chronic kidney disease. These findings are particularly promising, given the lack of effective treatments for fibrotic diseases, which often lead to progressive organ dysfunction and ultimately end-stage organ failure. The potential of PFD to halt or reverse fibrosis represents a major advance in the development of new therapeutic strategies for these devastating diseases.

Moreover, the improved renal function after IRI coincided with a significant decrease in collagen deposition in the kidney. We measured Col1A1 gene expression in kidney extracts. Col1A1 is involved in the production of type 1 collagen. We observed a considerable reduction in Col1A1 mRNA expression under the effect of PFD. This result concurs with the findings of Shimizu et al., who reported that PFD administration was associated with the suppression of the mRNA expression of the genes that code for type I collagen and type IV collagen [28]. In this same direction, we measured the expression of TGF-beta and found a reduction in its mRNA level in the kidney. It is well known that TGF-beta is a growth factor with significant profibrotic effects.

Furthermore, TGF-beta has been shown to have a definitive role in the development of renal fibrosis in animal and clinical studies [29]. This cytokine is the direct target of PFD, and we found a decrease in the expression of this gene. Our results suggest that this blockage could be associated with improvements in renal function and a reduction in IRI-associated damage. Through a histopathologic analysis, we observed that fibrosis was severely reduced due to the oral use of PFD. Our findings regarding the antifibrotic properties of PFD are transparent and provide insight into the mechanism of the renoprotective effects of PFD.

TGF beta is a protein that plays a key role in regulating various cellular processes, including cell growth, differentiation, and immune function. The expression of TGF beta is controlled by several factors, including mRNA levels [30]. When the mRNA level of TGF beta decreases, TGF beta protein production may be reduced. However, a decrease in mRNA levels may not always lead to a significant decrease in protein levels. This is because the expression of proteins is regulated by many factors, including post-transcriptional modifications, protein stability, and degradation. Additionally, it is possible that the observed effects of pirfenidone on the animal model are mediated by other pathways that are independent of TGF beta regulation. Therefore, although the decrease in mRNA levels of TGF beta may not be significant, the observed effects of pirfenidone on the animal model suggest that it may be influencing TGF beta-related pathways in other ways. More research is needed to fully understand the mechanisms of action of pirfenidone and how it affects TGF beta expression and function.

Finally, we inquired about the anti-inflammatory effect of PFD in the chronic IRI model. Here, we showed a reduction in IL-18 levels, suggesting an improvement in the inflammatory reaction that triggers IRI progression. By decreasing inflammation, kidney fibrosis consequently decreases as well. This effect of PFD was already demonstrated in other studies through the measurement of cytokines (IL-1b, IL18, and others such as interleukin 2) and tumor necrosis factor-alpha levels.

One limitation of our study is that we used a relatively small sample size, which may have reduced the power to detect significant effects and increase the risk of type II errors. Another limitation is that we focused exclusively on male rats and did not investigate potential sex differences in the effects observed. Furthermore, the use of a specific animal model may limit the generalizability of our findings to other species or disease models. Future studies with larger and more diverse samples, as well as

investigation of potential sex differences and alternative animal models, are needed to further explore the mechanisms and potential translational applications of our findings.

## 4. Materials and Methods

### 4.1. Sample Size

The short-term efficacy of pirfenidone upon ischemia–reperfusion injury was rated at around 92.31%. Only 7.69% of the subjects developed fibrosis and a chronic presentation. These data are based on the results obtained by Qiu et al. [19], with a power of 80%, a ratio of controls to cases of 1, 92.7% of the controls exposed and 7.60% of the cases exposed. A total of 18 animals was determined as necessary for the study by calculating the sample size aided by the EPI-INFO software 7 Atlanta, GA, USA.

### 4.2. Ethical Statement

The methods section was defined following the ARRIVE guidelines for reporting animal research [31]. This study was performed in compliance with Mexican standards that dictate the technical specifications for the production, care, and use of laboratory animals (062-NOM-ZOO-1999) and the NIH Guidelines for the care and use of laboratory animals. The Local Institutional Ethics Committee of Research in Health at "Instituto Mexicano del Seguro Social" approved the study protocol under registration number R-2019-1301-016.

### 4.3. Experimental Animals

The experimental model was established using 18 male Wistar rats per group. The animals were housed under the strict care and handling of an experienced veterinarian in the Centro de Investigacion Biomedica de Occidente (CIBO) animal facilities. The animals had access to water and standard rodent food ad libitum. Appropriate habitat conditions were supplied, respecting 12–12 h of light/dark cycles. Temperatures and humidity were controlled according to the animals' specific needs. At the end, the rats were humanely euthanized as described within the NIH guidelines for the care of laboratory animals.

### 4.4. Study Design

The study design is described in Figure 6. In summary, eighteen male Wistar rats, weighing approximately 300–350 g, were randomly organized into three experimental groups: (1) Sham ($n = 6$), (2) Unilateral IRI ($n = 6$), and (3) Left unilateral IRI + PFD ($n = 6$). Group 3 started the PFD treatment (0.5 mg/kg) 24 h before the IRI procedure, followed by intragastric administration of PFD (0.5 mg/kg/day) for 30 days. PFD was provided by Cellpharma. Allocation, analysis follow-up, and randomization of the groups are shown in Figure 6.

### 4.5. Surgical Experimental Procedures

All animals were anesthetized using xylazine (8 mg/kg, i.p., Richmond Vet Pharma) for sedation and analgesia and ketamine (100 mg/kg, i.p, PiSA Pharmaceutics) for maintaining anesthesia. The left renal hilum was dissected after making an incision in the abdominal medium line to expose the kidney pedicle. In the IRI groups (2 and 3), the circulation was occluded with a microvascular clamp for 45 min [32,33]. After the ischemia period, the renal hilum clamp was released, and the surgical wound was closed according to the precise anatomical planes in the sham group. The same procedure, without the occlusion, was performed in the sham group (Figure 6). After surgery, the animals were kept under observation in an incubator at 37 °C until their full recovery. Then, the animals were placed in a cage for 30 days for reperfusion (Figure 6). In the PFD-treated group, the animals received an orogastric dose of PFD (0.5 mg/kg) 24 h before surgery, followed by a single daily dose of PFD (0.5 mg/kg/day) administered orally at the same time of the day for the 30 days of the reperfusion. At the end of the reperfusion, all animals were anesthetized using xylazine (8 mg/kg, i.p.) and ketamine (100 mg/kg, i.p.) to collect their blood and both kidneys' cortex for mRNA extraction as well as the whole kidneys for histological analysis.

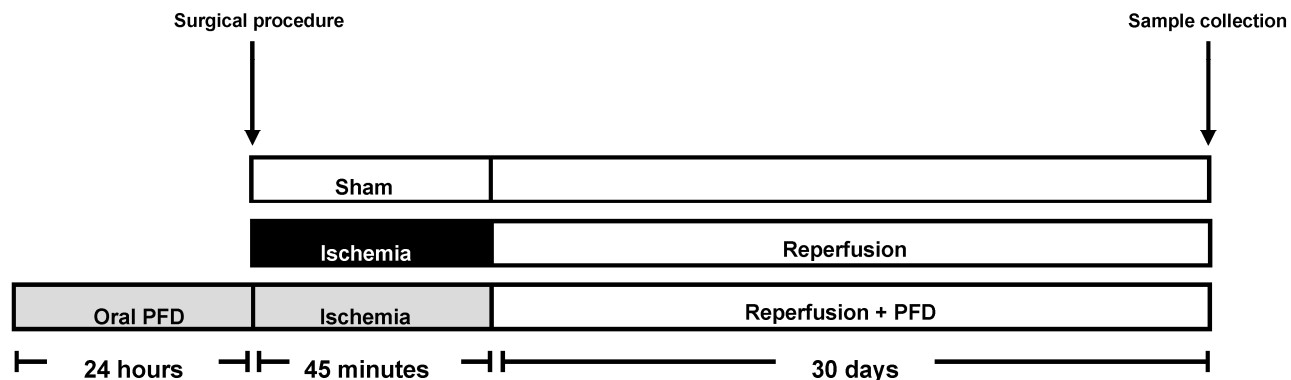

**Figure 6.** Experimental outline. Six male Wistar rats were treated with 0.5 mg/kg/day of PFD orally administrated (light gray bar) 24 h before the left renal hilum was blocked for 45 min (ischemia), followed by 30 days of reperfusion. The administration of a daily oral dose of PFD (0.5 mg/kg/day) was continuous throughout the 30 days. A simulated surgery sham group (*n* = 6, white bar) and an IRI without PFD group (*n* = 6, black bar) that followed the same IRI surgical protocol were also analyzed. The gray scale identification is maintained in all figures. Blood and renal tissue samples were collected at the end of the 30 days in the three groups.

### 4.6. Real-Time PCR

The whole kidney was used to obtain total RNA with RiboZol RNA extraction reagent (AMRESCO, VWR life sciences, Radnor, PA, USA). Reverse transcription was achieved with the QuantiTect Reverse Transcription Kit (Qiagen, Hilden, Germany). Quantitative real-time PCR was performed using the ready-to-use mix of enzyme and SYBR Green qEva-Green (qARTA Bio, Inc., Carson, CA, USA) in a Lightcycler 96 instrument (Roche Molecular Systems, Inc., Pleasanton, CA, USA). The expression of kidney injury molecule 1 (KIM-1), Collagen 1A1 (Col1A1), and TGF-beta was measured using the following specific oligonucleotides: F-TCCTGTGGGATTCATGCAGT and R-GCAGGAGGCCTGAAATGAAG for KIM-1, F- AAGGCTCCCCTGGAAGAGAT and R-CAGGATCGGAACCTTCGCTT for Col1A1, and F-AGGAGACGGAATACAGGGCT and R-CCACGTAGTAGACGATGGGC for TGF-beta. Briefly, the amplification conditions were: 10 s at 95 °C, 30 s at primer-specific annealing temperature and 30 s at 72 °C for 40 cycles. The relative gene expression was quantified using the $2^{-\Delta\Delta Ct}$ threshold method. Hypoxanthine phosphoribosyltransferase (HPRT) mRNA was used as a housekeeping gene.

### 4.7. Luminex Assay

To evaluate the inflammatory cytokines, we used a rat premixed magnetic bead-based Luminex assay kit (R&D Systems, Minneapolis, MN, USA) to detect IL-18 in the serum following the fabricant's instructions. The Luminex assay was performed using a Luminex 200 (Luminex Corporation, Austin, TX, USA) instrument.

### 4.8. Histological Staining

Before the staining procedures, the kidneys were fixed in 10% buffered formalin, embedded in paraffin blocks, and cut into 10 μm sections. The kidney sections were stained with hematoxylin and eosin (H&E), periodic acid–Schiff (PAS) stain, Masson's trichrome, and Sirius red. Photographs were acquired in an optical microscope using a 20× objective, and ten pictures of each section were randomly taken for histological analysis. The percentage of blue Masson trichrome-stained tubulointerstitial area and scale was measured using ImageJ v.1.50i Software (NIH, Bethesda, MD, USA).

### 4.9. Statistical Analysis

Data are presented as mean ± standard error of the mean (SE). The analysis was performed using the Shapiro–Wilk test for normality. One-way RM-ANOVA was applied

for a comparison between groups. Sigma Stat was used to analyze the data, and Sigma plot software v.12 (Systat Software, San Jose, CA, USA) was used to create the graphs. The threshold for statistical significance was set at $p < 0.05$.

## 5. Conclusions

Our findings indicate that PFD may have a protective effect against chronic renal ischemia–reperfusion injury by decreasing the development of fibrosis and inflammation. This research offers novel perspectives for the treatment and control of chronic renal impairment that arises after IRI. The potential of PFD to reduce fibrosis and inflammation in chronic renal IRI represents a significant advance in the field of nephrology, as it opens new avenues for the treatment and control of chronic renal impairment. By inhibiting the profibrotic and pro-inflammatory signals associated with fibrosis, PFD may help to prevent the progression of renal disease and preserve kidney function. These findings highlight the importance of additional research to fully understand the mechanisms of action of PFD in renal fibrosis and explore its potential as a therapeutic agent for other fibrotic diseases.

**Author Contributions:** Conceptualization, A.F.-A., L.C.-S. and A.G.-O.; Methodology, J.P., A.F.-A., D.G., R.G.-G., M.G.V.-M. and B.M.-C.; Validation, A.G.-O.; Formal analysis, M.G.V.-M., C.F.-O., F.J.B.-C. and E.A.R.-E.; Investigation, J.P., L.C.-S. and A.G.-O.; Writing—original draft, Z.M. and B.M.-C.; Writing—review & editing, Z.M., M.G.V.-M., E.P.-d.B., B.M.-C and A.G.-O.; Visualization, E.P.-d.B., C.F.-O., F.J.B.-C., E.A.R.-E. and L.C.-S.; Supervision, Z.M. All authors have read and agreed to the published version of the manuscript.

**Funding:** Z.M. and R.E. were supported by the project 653 "Investigadores por Mexico" program of CONACYT Mexico. The project is part of a protocol granted by CONACYT "Ciencia de Frontera" 345366 FORDECYT-PRONACES to Z.M. and AFDICT 230792 to E.P.-d.B.

**Institutional Review Board Statement:** The animal study protocol was approved by the Institutional Review Board of the Hospital de Especialidades, Instituto Mexicano del Seguro Social under the registration number R-2019-1301-016 for studies involving animals.

**Informed Consent Statement:** Not applicable.

**Data Availability Statement:** The data presented in this study are available on request from the corresponding author. The data are not publicly available due to confidentiality concerns or restrictions imposed by the institution.

**Acknowledgments:** The authors thank the CIBO-IMSS staff for the technical support in the experimentation process. They especially thank the "Fundación IMSS" for all the administrative support.

**Conflicts of Interest:** The authors declare no conflict of interest.

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
