# Peer review of "Pharmacological Blockade of TGF-Beta Reduces Renal Interstitial Fibrosis in a Chronic Ischemia–Reperfusion Animal Model"

_ddc, doi:10.3390/ddc2010009_

Round 1

Reviewer 1 Report

This is an interesting study and the authors have collected a unique dataset using cutting edge methodology and well-designed series of experiments. Overall, this is a clear, concise, and well-written manuscript. The introduction is relevant and rationale-based. Sufficient information about the previous study findings is presented for readers to follow the present study rationale and procedures. The methods are generally appropriate, although clarification of a few details (mentioned below) and provision of a rationale for the use of this particular method should be provided. Overall, the results are clear. In addition and in my opinion the paper has some shortcomings in regards to some data analyses/statistics.

I- Major comments:

1) In figure 4A; Authors please check again. How IL-1beta is totally undetected in IRI+PDF group!!

2) I believe the figures were created using GraphPad Prism. Please credit this software in the "Statistical analysis" part.

3)  In your experiments sets, One-Way ANOVA is a ideal test for your statistical analyses. Using of Student t-test may is totally not accurate with your sets.

4) Are you sure your data are mean +/- SEM and not mean +/- SD?? The SEM looks huge in all of your results and figures!

II- Minor comments:

- Creatinine should be serum creatinine in Figure 1

- 157.51pg/mL ±58.436 should add the units and %  in other text after the SD.

- It is IL-1beta not IL-1b

Author Response

Reply to reviewer 1.

This is an interesting study and the authors have collected a unique dataset using cutting edge methodology and well-designed series of experiments. Overall, this is a clear, concise, and well-written manuscript. The introduction is relevant and rationale-based. Sufficient information about the previous study findings is presented for readers to follow the present study rationale and procedures. The methods are generally appropriate, although clarification of a few details (mentioned below) and provision of a rationale for the use of this particular method should be provided. Overall, the results are clear. In addition and in my opinion the paper has some shortcomings in regards to some data analyses/statistics.

R/ Thank you for your valuable feedback and positive remarks on our manuscript. We are grateful for the time and effort you dedicated to evaluating our work. Based on your suggestions, we have revised our initial version of the manuscript, addressing each point separately.

Each point made by you is addressed below, explaining what we have changed and where the changes are to be found in the manuscript. All the changes are highlighted in the revised manuscript.

I- Major comments:

1) In figure 4A; Authors please check again. How IL-1beta is totally undetected in IRI+PDF group!!

R/ At first, we attributed the result to the detection limit of the test. Nevertheless, following the reviewer’s suggestion, we removed the figure from the manuscript.

2) I believe the figures were created using GraphPad Prism. Please credit this software in the "Statistical analysis" part.

R/ We appreciate the reminder about the credits. All figures were made using Sigma Plot. We included that information in the methods section.

3)  In your experiments sets, One-Way ANOVA is a ideal test for your statistical analyses. Using of Student t-test may is totally not accurate with your sets.

R/ We agreed with the reviewer, when conducting an analysis of a three-arm experimental design, a common approach to address multiple comparisons is to use a multivariate analysis of variance (MANOVA) or a repeated measures analysis of variance (RM-ANOVA). Both methods can account for the correlations among the repeated measures and provide a more robust test for multiple comparisons than using multiple univariate tests, such as Student's t-test, which can result in an excess of false positive findings.

Then, we checked all the statistical analysis using One-way RM-ANOVA. We will only highlight the significant differences in the comparison groups important for the conclusions of the study.

4) Are you sure your data are mean +/- SEM and not mean +/- SD?? The SEM looks huge in all of your results and figures!

R/ To response to this question, we added all individual measurements in the figures as dot plot overlying the bars.

 II- Minor comments:

- Creatinine should be serum creatinine in Figure 1

R/ Added

- 157.51pg/mL ±58.436 should add the units and %  in other text after the SD.

R/ Done

- It is IL-1beta not IL-1b

R/ As a reviewer suggestion, this information was deleted.

Reviewer 2 Report

The results of the presented work suggest an interesting potential clinical use to reduce long-term fibrosis complications in kidneys subjected to prolonged hypoperfusion. The presented is study is interesting, the animal experiment is well designed, however several concerns arise starting from sample processing to statisticals as follwos:

1) English grammar check-up is needed, "Spanglish" sentence structures need to be revied.

2) Gene expression and protein analysis from the injured kidney tissue after I/R should dissect cortex and medulla, and evaluated separately not only as the whole kidney.

3) Data presentation should avoid bar charts. Please change to dot plots, showing each individual data! Also, Fig1C in the present form is inadequate, you have to show all individual body weight data for each group.

4) Fig3 (histology) should show representative pictures of similar kidney area from each group. Now the IRI pictures show solely tubular structure (presumably distal tubules) but the control and IRI+PFD kidney photos are taken from periglomerular area. Please correct. Also, a scale bar for each photo is missing. PAS stain pictures give no reasonable information over trichrome, could be discarded (anyway are poor quality).

5) The statistical comparison of 3 experimental groups using T-test or Mann-Whitney is wrong. Please re-calculate all statistics using Kruskal-Wallis and an appropriate post-hoc test.

6) Fig 4B makes no sense, as there are no data in sham and treated group, therefore statistical evaluation can't be done either. Pls delete this figure.

7) As marker of renal injury, only serum Crea and renal KIM-1 gene expression was measured. Please include at least an Lcn2 gene expression and a serum + urine Lcn2 (NGAL) protein measurement, that has better correlation to tubular injury.

8) TGF-beta mRNA expression does not obviously correlate with active TGFb protein levels in the kidney. Therefore, a TGFb immunoblotting is a minimal need, or at least a TGFbeta immunohistochemistry. Also, when fibrosis markers are evaluated, the dysregulation of MMP/TIMP system has to be evaluated as well at gene expression and protein level (or at least using a gelatinase zymography).

9) Similarly, the referee does not understand why only 2 serum inflammatory markers were measured, despite the cost-effective qPCR analysis for several renal inflammatory markers (CCL2, IL1b, IL6, C3, C4b, etc) is possible (IL1b for instance in renal tissue for sure will give data for statistics, not like the serum levels).

Reviewer 4 Report

The authors investigated the effect of PFD on fibrosis in a chronic IRI rat model. They evaluated the influence of PFD on the expression of profibrotic molecules and collagen deposition in renal tissues. The theme is very relevant and important for the research area. The work is well-written and well-organized. The research is quite relevant and the results are promising. I only make a suggestion to specify the producers of the medicines used (PFD, xylazine, ketamine).

Round 2

Reviewer 2 Report

Some important concerns have been addressed by the authors, like grammar corrections, changesof data presentation and statistical analysis. 

Still, the quality of photomicrographs remains low, hopefully authors can provide large images at 300dpi resolution. Additionally, the authors did not complete the small amount of requested qPCR studies to verify inflammatory and renal damage effects, weakening the results. Also, TGFb protein expression data are lacking so non-significant TGFb mRNA changes need to be discussed as a study limitation - increase in TGFb gene expression does not mean active TGFb protein overexpression.

Author Response

Reply responses to reviewer 2

Still, the quality of photomicrographs remains low, hopefully authors can provide large images at 300dpi resolution.

The reviewer is right, when we switch to PDF the quality of the photographs drops, so we can send the original archives (.TIFF) for the management of the journal.

Additionally, the authors did not complete the small amount of requested qPCR studies to verify inflammatory and renal damage effects, weakening the results.

We appreciate the concern of the reviewer, we understand that with more results the impact of the manuscript would improve. Nevertheless, due to limitations in time, funding, and resources, it was not possible to complete all the qPCRs that were requested by the reviewer. We will take the reviewer's feedback into consideration for future studies related.

Also, TGFb protein expression data are lacking so non-significant TGFb mRNA changes need to be discussed as a study limitation - increase in TGFb gene expression does not mean active TGFb protein overexpression

R/ We added the information below to “Discussion” section.

TGF beta is a protein that plays a key role in regulating various cellular processes, including cell growth, differentiation, and immune function. The expression of TGF beta is controlled by several factors, including mRNA levels. When mRNA levels of TGF beta decrease, it can result in reduced TGF beta protein production. However, the decrease in mRNA levels may not always lead to a significant decrease in protein levels. This is because the expression of proteins is regulated by many factors, including post-transcriptional modifications, protein stability, and degradation. Additionally, it is possible that the observed effects of pirfenidone on the animal model are mediated by other pathways that are independent of TGF beta regulation. Therefore, although the decrease in mRNA levels of TGF beta may not be significant, the observed effects of pirfenidone on the animal model suggest that it may be influencing TGF beta-related pathways in other ways. More research is needed to fully understand the mechanisms of action of pirfenidone and how it affects TGF beta expression and function.